# Omics-Based Platforms: Current Status and Potential Use for Cholangiocarcinoma

**DOI:** 10.3390/biom10101377

**Published:** 2020-09-28

**Authors:** Yu-Chan Chang, Ming-Huang Chen, Chun-Nan Yeh, Michael Hsiao

**Affiliations:** 1Department of Biomedical Imaging and Radiological Sciences, National Yang-Ming University, Taipei 112, Taiwan; jameskobe0@gmail.com; 2Genomics Research Center, Academia Sinica, 128 Academia Rd., Sec. 2, Nankang-Dist., Taipei 115, Taiwan; 3School of Medicine, National Yang-Ming University, Taipei 112, Taiwan; mhchen9@gmail.com; 4Center of Immuno-Oncology, Department of Oncology, Taipei Veterans General Hospital, Taipei 112, Taiwan; 5Department of General Surgery, Liver Research Center, Chang Gung Memorial Hospital, Linkou, Taoyuan 333, Taiwan; yehchunnan@gmail.com; 6Cancer Genome Research Center, Chang Gung Memorial Hospital, Linkou, Taoyuan 333, Taiwan

**Keywords:** cholangiocarcinoma, genomic alterations, omics, metabolites, small molecule compounds/drugs

## Abstract

Cholangiocarcinoma (CCA) has been identified as a highly malignant cancer that can be transformed from epithelial cells of the bile duct, including intrahepatic, perihilar and extrahepatic. High-resolution imaging tools (abdominal ultrasound, computed tomography and percutaneous transhepatic cholangial drainage) are recruited for diagnosis. However, the lack of early diagnostic biomarkers and treatment evaluation can lead to serious outcomes and poor prognosis (i.e., CA19-9, MUC5AC). In recent years, scientists have established a large number of omics profiles to reveal underlying mechanisms and networks (i.e., IL-6/STAT3, NOTCH). With these results, we achieved several genomic alteration events (i.e., TP53_mut,_ KRAS_mut_) and epigenetic modifications (i.e., DNA methylation, histone modification) in CCA cells and clinical patients. Moreover, we reviewed candidate gene (such as NF-kB, YAP1) that drive gene transcription factors and canonical pathways through transcriptomics profiles (including microarrays and next-generation sequencing). In addition, the proteomics database also indicates which molecules and their directly binding status could trigger dysfunction signatures in tumorigenesis (carbohydrate antigen 19-9, mucins). Most importantly, we collected metabolomics datasets and pivotal metabolites. These results reflect the pharmacotherapeutic options and evaluate pharmacokinetic/pharmacodynamics in vitro and in vivo. We reversed the panels and selected many potentially small compounds from the connectivity map and L1000CDS^2^ system. In this paper, we summarize the prognostic value of each candidate gene and correlate this information with clinical events in CCA. This review can serve as a reference for further research to clearly investigate the complex characteristics of CCA, which may lead to better prognosis, drug repurposing and treatment strategies.

## 1. Introduction to CCA

Cholangiocarcinoma (CCA) is a malignant tumor transformed from bile duct epithelial cells [1,2]. According to the anatomic location, it is divided into three different subtypes—intrahepatic (iCCA, 10–20%), perihilar (pCCA, 50–60%) and extrahepatic (eCCA, 20–30%) [3]. iCCA is the second most common primary liver cancer after hepatocellular carcinoma. Recently, iCCA was divided into two subtypes (small- and large-duct types) based on the production of mucin and phenotype by immunohistochemically evaluated. The small-duct-type cancer usually has a history of chronic liver disease (five-year overall survival is longer ~60%, most of which are genetic changes of IDH1, BAP1 and FGFR2) and large-duct iCCA and pCCA have many clinicopathological characteristics and are accompanied by premalignant biliary intraepithelial neoplasia (BiIIN) in adjacent ducts (five-year overall survival is shorter, accounting for about 20%, most of which are genetic changes of SMAD4 and KRAS7) [4]. The incidence of CCA has been increasing worldwide, especially in Asia [5]. Analyzing the annual mortality rate of CCA, the average mortality rate for the period 2010–2014 was 0.3–2.88 (per 1 × 10^5^ inhabitants) [6]. However, higher mortality rates (>4, per 1 × 10^5^ inhabitants) including China [7], Japan [8], South Korea [9], Thailand [10], Taiwan [11] and Austria [12]. It has also been confirmed that men have a higher odds ratio than women [13]. Surgical resection is the only potentially curative clinical therapy and results in a five-year overall survival rate between 15–40% [14,15,16,17,18]. More than two-thirds of patients relapse after resection. The main reasons for the high recurrence rate are positive surgical margins, lymph node metastasis, vascular infiltration and multiple tumor regions.

Recent reports on CCA have identified several potential risk factors and carcinogens. Biliary tract infections caused by parasitic infections can cause liver flukes (*Clonorchis sinensis* and *Opisthorchis viverrini*) and they have the highest incidence of CCA in Northeastern Thailand [19]. Primary sclerosing cholangitis, hepatitis B/C virus (HBV/HCV) infection, biliary stone diseases, congenital biliary malformations, cirrhosis and exposure to aromatic toxins are all potential CCA disease etiologies [20]. In addition, smoking, alcohol consumption and diabetes also contribute to the risks of CCA. In particular, 1,2-dichloropropane (1,2-DCP) has been demonstrated as a carcinogen for CCA. The main role of 1,2-DCP is interference with CYP450-dependent proliferation and apoptosis [21]. For treatment, palliative chemotherapy with gemcitabine and cisplatin is the standard 1st line treatment for unresectable or relapse/recurrence CCA patients. However, the response to such therapy remains limited [22]. Most importantly, CCA can escape several known tolerance mechanisms induced by chemotherapy (including cisplatin, gemcitabine and other combinations) [23]. Drug development that relies on well-known signal transductions needs in-depth discussion.

Compared with known pathogenic factors and diagnosis protocols, the oncogenic mechanisms and treatment strategies still need to be improved and investigated [24]. Therefore, this review summarizes these omics-based datasets. These comprehensive approaches aim to solve the bottleneck of CCA study.

## 2. Genomics Alterations and Epigenetic Changes in CCA

In recent years, 38 CCA tumors have been added to The Cancer Genome Atlas (TCGA). Through these changes, we can achieve alterations of tumor suppressor genes and oncogenes. TCGA has established pipelines through the following platforms: whole exome sequencing, affymetrix SNP 6.0 copy number array, RNA sequencing (RNA-seq) containing microRNA/long noncoding RNA, DNA methylation and reverse-phase protein array (RPPA). Combining this evidence, scientists have revealed genetic alterations and epigenetic events in CCA [25,26].

### 2.1. Genomic Alterations

The genome sequencing study conducted by Nakamura et al. in CCA identified many driver gene changes. A total of 239 pairs of CCA and corresponding normal tissues (including 137 iCCA, 74 eCCA and 28 gallbladder cancers) were sequenced by whole exons [27]. The results show that FDFR2 was specifically fused with iCCA (6/109, 55%), and the two FGFR2 fusion proteins can induce ligand-independent FGFR2 autophosphorylation and downstream mitogen-activated protein kinase signaling (MAPK). After activation, anchorage-independent growth and tumor formation can be increased. On the other hand, in eCCA cases, cyclic AMP (cAMP)-dependent protein kinase (PKA) signaling components can form new fusions ATP1B-PRKACA and ATP1B-PRKACB (25/260, 9.6%). These fusions also increase the PKA activity and activate MAPK signaling [28]. Aside from somatic alterations, they also identified several driver gene landscapes in CCA. Sixteen putative tumor-suppressor genes have been recognized, including *TP53*, *ARID1A*, *BAP1*, *ARID2*, *PBRM1*, *APC*, *EPHA2*, *ELF3*, *ATM*, *BRCA2*, *RPL22*, *ACVR2A*, *STK11*, *MLL3 (KMT2C)*, *NF1* and *MLL2 (KMT2D)*. After annotation and validation, they finally announced that the key factors in CCA include kinases (*FGFR1*, *FGFR2*, *FGFR3*, *PIK3CA*, *ALK*, *EGFR*, *ERBB2*, *BRAF* and *AKT3*), oncogenes (IDH1, IDH2, CCND1, CCND3 and MDM2) and tumor suppressor genes (*BRCA1* and *BRCA2*) [27]. Tian et al. also identified a comprehensive genomic profile of CCA in China’s population. In this study, they inputted 44 iCCA and 22 eCCA cases. The results show: TP53 (62.1%), KRAS (36.4%), SMAD4 (24.2%), TERT (21.2%), ARID1A (19.7%), CDKN2A (19.7%), KMT2C (9.1%), RBM10 (9.1%), ERBB2 (7.6%) and BRAF (7.6%). Similarly, they observed specificities expressed in iCCA (STK11, CCND1, FGF19, FGF3, FGF4 and PBRM1). For FGFR2, they are expressed in both Western and Chinese cohorts [7]. Through the cBioPortal website [29,30], we merged six clinical cohorts to analyze their genetic alteration events, survival time and the proportion of several demographic factors (Figure 1). We also summarized the top frequency copy-number alteration (CNA)/mutation/fusion gene, as shown in Table 1. In addition, we performed a graph to indicate the multiple genetic alteration events in above studies (Figure 2).

### 2.2. Epigenetic Events

In an epigenetic study, there are often several cell cycle-related factors and genes involved in DNA repair functions that are frequently methylated in CCA. Therefore, dysfunction of these genes may suppress tumor suppression and unusual cell behavior. *MLH1*, *DCLK1*, *CDO1*, *ZSCAN18*, *ZFG331*, *p14*, *p16*, *DAPK*, *CCND2*, *CDH13*, *GRIN2B*, *RUNX3*, *TWIST1*, *EGFR* and *LKB1* are most of all CpG sites that have exhibited hypermethylation and then loss of function in CCA tumorigenesis [31,32,33,34,35,36,37,38,39,40]. Moreover, *RASSF1A*, *FHIT*, *14-3-3*, *TMS1*, *CDH1*, *CHFR*, *GSTP*, *TIMP3*, *SEMA3B*, *RAR-β*, *MGMT*, *BLU*, *THBS1*, *RIZ1*, *OPCML* and *PTGS2* have also been mentioned in CCA subtypes or cell lines. In histone modification, CCA studies have shown that histone methylation, acetylation and chromatin remodeling are unbalanced. H3K9, H3K27 and H4K20 methylation leads to transcriptional repression, while H3K4 methylation leads to transcriptional activation [41]. In an in vivo CRISPR model, mutation of Arid1a leads to SWI/SNF chromatin complex remodeling. Moreover, both of the SWI/SNF family members, AT-rich interaction domain (ARID) 1A and polybromo 1 (PBRM1) showed loss of function in CCA tumorigenesis [42,43]. Non-synonymous mutations in chromatin remodelers are typically distributed throughout the gene. There are no definable hotspots, and the significance of many mutations is unknown. Therefore, histone modification and chromatin remodeling still require in-depth investigation and innovative research.

An important aspect of epigenomics is microRNAs (miRNAs). Deregulation of miRNAs can induce the initiation and progression of cancers by modifying target tumor suppressor genes or oncogenes. Recently, it has been reported that IL-6 regulates the activity of DNA methyltransferase 1 (DNMT1) by miRNAs in CCA cells [44]. DNMT1 can cause comprehensive changes in the degree of methylation. In CCA, various genes have been reported to have frequent methylation, including *MLH1*, *DCLK1*, *CDO1*, *ZSCAN18*, *ZNF331*, *p14*, *p16*, *DAPK*, *CCND2*, *CDH13*, *GRIN2B*, *RUNX3*, *TWIST1*, *EGFR* and *LKB1* [45]. After verification, it was confirmed that the miR-148a/miR-152 family directly binds to the DNMT gene. Interestingly, DNMT1 regulates the hypermethylation of its CpG islands and then loses these tumor suppressor functions. Therefore, they form a negative feedback regulatory loop between DNMT1 and miR-148a/miR-152 family in tumorigenesis. Moreover, several tumor suppressor miRNAs have been shown in the CCA model. miR-370 was inhibited following global hypermethylation. Then, downstream target mitogen-activated protein kinase 8 (MAP3K8) was regulated by miR-370 [46]. Another study also showed that miR-376c was regulated by DNA methylation and associated with tumor suppression by targeting growth factor receptor-bound protein 2 (GRB2) [47]. Yang et al. observed that miR-144 was reduced in CCA tissues. They also verified platelet-activating factor acetylhydrolase isoform 1b (LIS1) as the direct target in CCA [48]. Although the target gene of direct action of some miRNAs has not yet been determined, through statistics and calculations, we list the most significant changes of miRNAs in CCA, especially between normal adjacent tissues and tumor groups. These miRNAs include oncomiRs (miR-183, miR-96, miR-182 and miR-181b) and tumor suppressor miRs (miR-99a, miR-125b-2, miR-621, miR-551b, miR-378c, miR-148a, miR-139, miR-378, miR-483, miR-885, miR-122, miR-490, miR-675 and miR-1258). Although many have not been examined and studied, these candidates have the potential for follow-up research. We drew a heat-map from the OncoMir Cancer Database (OMCD) to illustrate the expression of the most significantly different miRNAs between normal adjacent tissues and tumor sites according to appropriate criteria (Figure 3 and Table 2) [49]. Regarding the regulation of non-coding genes, not only mentioned the expression of microRNA in CCA, but also mentioned long-noncoding RNA (LncRNAs) and circular RNA (circRNAs) as contributors to the CCA model (Table 3). Zhang et al. revealed that LncRNA–colon cancer-associated transcript 1 (CCAT1) increased in tumor parts, and their result show that knockdown of CCAT1 suppressed the migration and invasion of iCCA cells. Then, they demonstrated the epithelial–mesenchymal transition (EMT) activation through the CCAT1-miR-152 axis in CCA [50]. Moreover, Xu et al. observed that CCAT2 had a similar function to CCAT1 in CCA. Overexpression of CCAT family members promotes CCA cell migration and invasion by EMT induction [51]. In addition, miR-590-3p was also revealed to suppress the EMT procedure through inhibition of the ZEB2 gene in iCCA [52]. On the other hand, LncRNA H19 (chromosome11p15.5) is also involved in cell proliferation, apoptosis and migration/invasion through EMT [53]. Li et al. also showed that lncRNA SOX2ot (SOX2 overlapping transcript) promotes cell proliferation and invasion in CCA cells [54]. Gao et al. confirmed that miR-10a-5p is one of the targets of PTEN and the miR-10a-5p inhibitor can suppress the PTEN expression and decrease inhibited Akt phosphorylation status in CCA cells. Therefore, miR-10a-5p regulates cholangiocarcinoma cell growth through the Akt pathway [55]. Similarly, more candidate miRNAs were reported over five years ago. Therefore, we do not mention them in detail in this review.

## 3. Canonical Pathways and Key Drivers of the Transcriptome

Several driver genes of CCA have been revealed through next-generation sequencing and their transcriptome datasets. Li et al. announced that Forkhead box M1 (FOXM1)/NF-kB has an inverse correlation between methionine adenosyltransferase 1(MAT1A) in both hepatocellular carcinoma and CCA (Pearson’s rho = −0.36, *p* = 1.24 × 10^−190^) [56]. MAT1A can mainly synthesize S- adenosylmethionine (SAMe) levels, which can prevent cholestatic liver injury. MAT1A has been identified that the expression level of MAT1A in primary and metastatic CCA is reduced compared with normal bile duct cells. Among them, MAT1A exerted a tumor suppressor effect and can also be used as an independent prognostic factor (Cox coefficient = −0.258, *p* = 6.8 × 10^−3^). In addition, prohibitin 1 (PHB1) plays a tumor-suppressive role and positively regulates MAT1A while suppressing c-Myc [57]. NAD(P)H-quinone oxidoreductase 1 (NQO1) plays an important role in chemo-resistance (5-fluorouracil and doxorubicin) and proliferation in several CCA cell lines. NQO1 enzyme activity can prevent potent cytotoxicity of chemotherapy drugs and eliminate cell apoptosis caused by caspase 3/7. [58]. Yeh et al. also claimed that TNNI3K has been noticed in array comparative genomic hybridization (aCGH) along with its overexpression in CCA cells and patients. TNNI3K is highly expressed and enhances cell growth and colony forming ability in vitro. [59]. Moreover, carbohydrate antigen 19-9 (CA19-9) is recognized as a well-known biomarker in pancreatic cancer and CCA [60]. In addition to internal genetic mutations and aberrant expression, external pathogens are also possible carcinogenic mechanisms. *Opisthorchis viverrini* is a specific pathogen of CCA. Aksorn et al. also recognized novel potential biomarkers that are associated with these events in CCA. These candidates include immunoglobulin heavy chain, translocated in liposarcoma (TLS) and visual system homeobox 2 (VSX2) [61]. There are many typical and classic markers in CCA, including *KRAS*, *TP53*, *PROM1*, *CTGF*, *VIM*, *DKK1*, *SOX2*, *SOX17*, *MUC1*, *PTEN*, *PTPN14*, *c-Met*, *EGFR*, *VEGR*, *CD44* and *CDH1* [62,63,64,65,66,67,68,69,70,71,72], but this review specifically focuses on novel evidence from the past five years.

## 4. Protein Levels and Post-Translational Modifications in CCA

Not only DNA status and RNA expression level change in CCA, but several protein molecules have also been raised in recent studies. For comprehensive and large-scale screening, scientists established proteomics-based approaches to address these unsolved issues. Liquid chromatography/mass spectrometry was used to examine the differential abundance of protein molecules in CCA. Myeloperoxidase (MPO), complement C3, inter-alpha-trypsin inhibitor heavy chain H4 (ITIH4), apolipoprotein B-100 and kininogen-1 isoform 2 (KNG1) increased in the tumor group compared to the benign group. On the other hand, trefoil factor 2, superoxide dismutase (SOD), kallikrein-1 (KLK1), carboxypeptidase B (CPDB) and trefoil factor 1 decreased in tumors [73]. Zhang et al. utilized two-dimensional difference in gel electrophoresis (2D-DIGE) combined with MALDI-TOF to identify mice exposed to 1,2-dichloropropane (1,2-DCP). The mice were exposed to 0, 50 and 250 ppm of 1,2-DCP through an inhalation exposure system (8 h/d, 7 d/week for four weeks) and the frozen livers was collected for further lysis and analysis. They captured several components include ferritin heavy chain (FTH1) (upregulated), ferritin light chain (FTL1) (upregulated), acyl-coenzyme a thioesterase 2 (ACOT2) (upregulated), GSTM1 (upregulated), aldehyde dehydrogenase X, mitochondrial (ALDH1B1), formimidoyltransferase cyclodeaminase (FTCD) (downregulated) and selenium-binding protein 2 (SELENBP2) (downregulated) [74]. Son et al. focused on eCCA specimens and collected bile samples from 18 patients with eCCA and the five patients with benign conditions for control group. By liquid chromatography–mass spectrometry (LC–MS) identified immunoglobulin kappa light chain, apolipoprotein E (APOE), albumin (ALB), apolipoprotein A-I (APOA1), antithrombin-III (SERPINC1), α1-antitrypsin (SERPINA1), serotransferrin (TF), immunoglobulin heavy constant mu (IGHM), immunoglobulin J chain (JCHAIN), complement C4-A (C4A) and complement C3 are all overexpressed compared with control groups [75].

There are several molecules with secretory ability, and it has recently been reported that various circulating RNAs and genes have been measured in serum. In the same situation, the establishment of secretome datasets also confirms that the extracellular molecules facilitate in CCA. These markers are also developing as prognostic/diagnostic factors in clinical study. Large et al. demonstrated the co-expression of plasma thrombospondin-2 (THBS2) with CA19-9 in CCA patients [76]. Furthermore, MMP7, IL-6, S100A6, DKK1 and SSP411 are also abnormally expressed proteins in CCA patients’ serum [70,77,78,79,80]. It is worth mentioning that Wang et al. declared that N-glycan status, including peak10 and NA3F2, presents a better diagnostic value than typical marker CA19-9 in eCCA patients’ serum [81].

## 5. Available Omics Datasets for CCA

Omics datasets have been applied in basic research studies. Da et al. merged the Gene Expression Omnibus (GEO) database GSE32225 with quantitative isobaric tags for relative and absolute quantification (iTRAQ) proteomics approaches to evaluate multiple candidate proteins associated with prognosis of CCA patients. In this study, they screened three candidate proteins APOF, ITGAV and CASK. They detected and scored the protein levels of all molecules on the CCA tissue microarray through IHC analysis. After comparison and statistical evaluation, they confirmed that peripheral plasma membrane protein CASK is considered the most significant molecule in this study [82]. The level of CASK expression has been determined to be correlated with overall survival (HR = 2.031, *p =* 0.027), vascular invasion *(p =* 0.009) and T classification *(p =* 0.017). Sulpice et al. established microarray chips (GSE45001) to predict that osteopontin is the independent predictor in iCCA and also performed gene expression profiling of tumor microenvironment in same cohort [83,84]. Peraldo Neia et al. revealed the gene and microRNA pattern of the iCCA patient-derived xenograft (PDX) model (GSE84918/GSE84938) [85]. Varamo analyzed the specific signatures to indicate the gemcitabine resistance CCA cell lines (GSE116118) [23]. Long-term exposure of gemcitabine-resistance cell lines (>1.5 uM, nine months) compared with sensible counterpart cells. They observed that several drug resistance- (ESR2 and FGF2) and EMT-related targets (COL5A2, RGS2, SNAI2, SPP1, TFPI2, TGFB1 and VIM) were regulated in the event. They further predicted several molecular mechanisms through 354 downregulated and 382 upregulated targets. These typical pathways include DNA replication, cell cycle and pyrimidine metabolism. In addition, they also found new molecules with differently expressed in their model (QPRT, RRM1, FOSB, PRSS1, etc.) [23]. Moreover, Gu et al. tried to find the linkage between microcystin–leucine–arginine (MC-LR) and iCCA (GSE140687). MC-LR has recognized that certain hepatotoxins can induce liver damage and liver cancer. MC-LR is also correlated with intrahepatic bile duct hyperplasia. Therefore, the group tried to explore the relationship between MC-LR, inflammation and intrahepatic bile duct epithelial cell proliferation. Although they have not yet come to the results of MC-LR in iCCA. However, they have established a microarray profile containing blank and MC-LR after 24 h of treatment in Huh28 cells. Meijer distinguished the circulating miR-16 and miR-877 as the diagnostic classifiers for eCCA (GSE117687) [86].

Recently, next-generation sequencing has been widely used. Yan et al. sequenced the single cell of iCCA and found intercellular crosstalk between iCCA cells and vascular cancer-associated fibroblasts (vCAFs) (GSE148773/GSE142784/GSE138709) [87]. They found that the IL-6/IL-6R axis was enriched in the interplay between vCAFs and iCCA cells. vCAFs contribute to the tumorigenesis and cancer stemness ability of iCCA through the axis. On the other hand, tumor exosomal miR-9-5p elicits IL-6 expression in vCAFs, which leads to iCCA epigenetic alterations. Saengboonmee et al. investigated the CCA cell lines in standard and high glucose medium (GSE137803). High glucose medium conditions promoted proliferation, metastatic and phosphorylation/ nuclear translocation of STAT3 [88]. Ma et al. claimed that HCC and iCCA have a varying degree of transcriptomics diversity through single-cell transcriptome profiling (GSE125449) [89]. Zhou et al. are performing ongoing non-coding RNA profiling to catalog the expression profiles of 1643 microRNAs in each sample (GSE93366). To determine the difference between serum and urine in CCA patients, Lapitz et al. established the array chips through a liquid biopsy approach (GSE144521) [90].

Many scientists have explored the aforementioned genetic alteration and epigenetic modification using omics-based technology. Bledea et al. dissected methylation data from IDH mutant CCA (GSE124617) [91], and their results support the suggestion that hypermethylation of enhancers associated with cell differentiation may mediate lack of differentiation in IDH1/2 mutant cancers. The profile of PTEN overexpression in CCA also has been checked by transcriptome array (GSE146196). A study classified their molecular panels and re-evaluated four classes for eCCA (KRAS: 36.7%, TP53:34.7%, ARID1A: 14% and SMAD4:10.7%) (GSE132305) [92]. All GSE datasets have been rationally designed, with appropriate criteria and comprehensively analyzed. Therefore, this information will become the foundation for future research.

## 6. Metabolomics Analysis of Oncometabolites and Metabolic Reprogramming in CCA

Isocitrate dehydrogenase 1 and 2 (IDH1 and IDH2) play a pivotal role in both DNA methylation and histone modification in CCA. IDH1 and IDH2 mutants occur only in ICC (10–28%). Wild-type IDH1 and IDH2 convert isocitrate to α-ketoglutarate (α-KG) in the cytoplasm and mitochondria, respectively. In contrast, mutant isoforms can produce anabolic 2-hydroxyglutarate (2-HG) [93,94]. As a result, many biologic processes that rely on α-KG, including ten-eleven translocase (TET) 2-mediated DNA demethylation, are disrupted. This leads to a wide range of epigenomic consequences, including increased 5-methylcytosine (5mC) content, decreased 5-hydroxymethylcytosine (5hmC) and increased histone H3 lysine 79 (H3K79) demethylation [95].

Liang et al. analyzed concentrated secretion ability through serum metabolomics establishment, and it was identified that the contents of LysoPC (14:0, C22H46NO7P) and LysoPC (15:0, C23H48NO7P) in the CCA group were reduced compared with the control groups [96]. Alsaleh et al. confirmed that lipid-derived compounds (phospholipids, bile acids and steroids) and amino acid products (phenylalanine) are associated with the risk of CCA in the United Kingdom [97]. The reduction of aberrant bile phospholipid secretion will cause the formation of “toxic bile” and exacerbate inflammatory bile duct damage. Chronic cholestasis and cholangitis may further lead to secondary cirrhosis of the liver and are susceptible to hepatobiliary cancer. In addition, this group also identified that purine metabolism and lipid metabolism have changed in the CCA urine metabolome in *Opisthorchis viverrini*-induced liver fluke-associated CCA [98]. Several comprehensive analyses merged with transcriptome, proteome and metabolome profiles of CCA have been published [99,100,101,102,103]. Combined with artificial intelligence, Urman et al. analyzed the metabolomics and proteomics analysis of bile from patients with benign and malignant tumors. Their results show that phosphatidylcholines (PCs), sphingomyelins (SMs) and ceramides (Cer) were found in serum and tissue through machine learning [104]. These metabolites and canonical metabolism signatures are all important indicators for detecting and evaluating diagnosis/prognosis/cytotoxicity.

## 7. Using Multiple-Omics Approaches to Determine Prognostic Factors

Zhou et al. detected that miR-378 is significantly associated with the TNM classification of malignant tumors (TNM) stage *(p =* 0.03) and lymph node metastasis *(p =* 0.018). Their results also reveal that miR-378 can serve as an independent prognostic predictor for CCA patients (Hazard Ratio (HR) = 1.735, 1.007–2.988, *p =* 0.041) [105]. In the CCA serum, Liu et al. also found that circulating miR-21 is a promising biomarker for diagnosis and a prognostic biomarker of CCA [106]. In serum of CCA patients, Silakit et al. also provided evidence that miR-192 expression level correlated with lymph node metastasis *(p =* 0.047) and poor survival (HR = 2.076, 1.004–4.291, *p =* 0.049) [107]. Wang et al. also described miR-26 in serum as a diagnostic and prognostic marker for CCA patients (overall survival *p =* 0.018 and progression-free survival *p =* 0.0381) [108].

There is an inverse correlation between suppressor of cytokine signaling 3 (SOC3) and tumor necrosis factor α-induced protein 3 (TNFAIP3) in CCA. Furthermore, patients with low SOC3 expression or high TNFAIP3 expression showed a dramatically lower overall survival rate. The SOC3 and TNFAIP3 expression levels had a significant prognostic power for CCA survival rate (SOC3: HR = 2.382, 1.253–4.722, *p =* 0.022, TNFAIP3: HR = 6.598, 2.584–14.261, *p =* 0.009) [109]. It has been reported that chitinase 3-like 1 (CHI3L1) glycoprotein is dysfunctional in CCA. Plasma CHI3L1 expression was particularly associated with poor survival in CCA patients (HR = 2.117, 1.027–4.363, *p =* 0.038, HR = 1.642, 0.780–3.455, *p =* 0.192) [110]. More important, we summarized the genes most related to survival in CCA (top 10) from the gene expression profiling interactive analysis (GEPIA) website [111]. Interestingly, we observed that no genes play a key role in overall survival and disease-free survival. These results may be divided into different systems due to proliferation/tumorigenesis and recurrence/metastasis in CCA (Table 4).

In our review, we have preliminarily sorted out the significant differences calculated on various analysis platforms in CCA based on statistical formulas, but these analyses are carried out in low-dimensional areas. The latest variable selection approach integrating multilevel omics data. Through horizontal/vertical integration model and various algorisms (penalized, Bayesian, etc.), we can obtain multidisciplinary data as the focus of future research [112,113]. These models will calculate and evaluate some of the above targets and combined with actual in vitro and in vivo studies to determine their authenticity. At current stage, we first collected significantly different genes in CCA through statistical models and predicted their participation in the interactions and networks by bioinformatics software (Figure 4).

## 8. Potential Small Molecule Compounds and Clinical Drugs from Omics Datasets

### 8.1. FGFR Family

Based on previous study, several specific inhibitors and small compounds were developed to combat aggressive features in CCA. Selective/non-selective small molecule kinase inhibitors were designed for FGFR gene fusion. Pemazyre (pemigatinib), which targets to fibroblast growth factor receptor 2 (FGFR2) rearrangement or fusion, was developed. This agent was proved to be effective by a multicenter, open-labeled single-arm trial in advanced cholangiocarcinoma patients and got FDA-approval recently [114]. Infigratinib passed Phase I and II studies of advanced CCA with FGFR2 gene fusion or other FGFR genetic alterations that failed platinum-based chemotherapy (NCT02150967) [115]. Erdafitinib (BALVERSA) is an oral pan-FGFR selective small molecule kinase inhibitor and the FDA granted accelerated approval for patients with locally advanced or metastatic urothelial carcinoma, with susceptible FGFR3 or FGFR2 genetic alterations. The initial reports of CCA pts with FGFR alterations treated with erdafitinib had encouraging efficacy and acceptable safety profile. (NCT02699606). ARQ-087 (derazantinib) is a small molecule kinase inhibitor with ATP competitiveness that can block the proliferative activity of cell lines with FGFR aberrations [116]. Its related Phase I/II trial also focused on iCCA patients with FGFR2 fusion (NCT01752920). Moreover, TAS-120 [117], AZD4547 [118] and CH5183284 (Debio-1347) [119] are undergoing early clinical trials for FGFR-driven tumors (NCT02052778, NCT01948297, respectively). Non-selective FGFR inhibitors ponatinib or pazopanib have also been used in specific populations (NCT02265341). In the case series study, ponatinib has been shown to reduce carbohydrate antigen 19-9 and decrease lymph node metastasis in chemotherapy-refractory CCA patients [120]. Finally, FGFR monoclonal antibodies have fewer off-target effects through competitive binding to FGFR2b and were evaluated in a Phase I trial (NCT02318329).

### 8.2. IDH Mutants

IDH1/IDH2-specific inhibitors are also important issues in the treatment of CCA. IDH mutations cause metabolic reprogramming and the production of oncometabolites. Epigenetic platforms are also affected by IDH mutation status. Therefore, AGI-5198 and AGI-6780 are divided into selective IDH1 and IDH2 inhibitors, respectively. In clinical trials (NCT02073994, NCT02273739), AG-120 (Ivosidenib) and AG-221 (enasidenib) were used as next-generation orally selective inhibitors (IDH1, IDH2, respectively). More important, it was determined that they can inhibit the production of oncometabolite 2-HG and impair the growth of IDH mutant cells.

### 8.3. Others

Through omics-based approaches, various genetic changes, oncogenic pathways and oncometabolites have been identified in CCA. Thus, more functional inhibitors have been designed for tumorigenesis: (1) The PKA inhibitor isoquinoline H89 can inhibit CCA cell proliferation by regulating the expression levels of PRKACA and PRKACB [121]; (2) S63845 and AZD1480 can regulate myeloid cell leukemia sequence 1 (Mcl-1), Janus kinase (JAK) and the signal transducer and activator of transcription (STAT) signaling cascade [122,123]; (3) KRAS is a typical mutant marker in CCA. However, there are no effective direct inhibitors of KRAS mutants and downstream pathways. In a clinical study, selumetinib (selective MEK 1/2 inhibitor) has a synergistic effect with gemcitabine and cisplatin in advanced CCA and ongoing clinical trials (NCT02042443, NCT01438554) [124]; (4) The combined treatment of CCA includes the PI3K/AKT/mTOR pathway (BKM120), the RAF–MEK–ERK axis with gemcitabine and platinum-based drugs [125]; (5) Tissue with abnormal expression of mesothelin in several malignant tumors has been observed. Amatuximab is a chimeric monoclonal antimesothelin antibody, while SS1P is a recombinant anti-mesothelin immunotoxin, and they are all involved in clinical trials of multiple cancer types [126]; (6) DNMT inhibitors azacitidine and decitabine and HDAC inhibitors vorinostat and romidepsin through rearrangement of chromatin remodeling form an epigenetic platform to rescue the function of tumor suppressor genes [127,128,129]; (7) The MET tyrosine kinase inhibitors tivantinib and cabozantinib were selected as chemotherapy strategies in clinical trials (NCT01954745) [130]; (8) reforafenib inhibited the Raf/Erk/Elk-1 pathway, and MALT1 has been identified as having a pivotal role in CCA tumorigenesis. Moreover, regorafenib has been recognized as the second standard treatment option in metastatic colon cancer. The clinical trial in CCA is currently at Phase II-III [131,132,133].

### 8.4. Immune Checkpoint Blockage

Immunotherapy was regarded as the next-generation strategy for cancer. Thus, immune checkpoint blockages are recruited and often used in combination therapy. The cytotoxic T lymphocyte associated protein 4 (CTLA-4) monoclonal antibody and the programmed death 1 (PD-1)/programed death ligand-1 (PD-L1) pathways are the spotlight of various cancers [134,135,136]. Investigation of iCCA specimens has shown that PD-L1 is highly expressed in tumor tissues and is inversely related to CD8+ T cell infiltration. The loss of HLA Class 1 antigen expression leads to CCA immune escape. On the other hand, the total mutation burden (TMB) and microsatellite instability (MSI) were reflected in the immune response in cancers. [137]. Therefore, patients with increased cases of high TMB or MSI-H may have a better immune response. Pembrolizumab, the PD-1 monoclonal antibody, was thawed for multiple treatment combinations (NCT02628067, NCT02268825). Merging all the information, we classified the clinical trial phase, targets and status of each potential compound (Table 5).

## 9. Future Prospects

CCA research still requires greater sample sizes and detailed clinical data as evidence and support. The omics-based dataset and techniques are suitable for CCA bottlenecks in basic research and clinical applications. Using the available information from RNA-sequencing, we can clearly classify subtypes and specific characteristics based on genetic/epigenetic background. Moreover, omics-based datasets also support a large number of expressions of each coding/non-coding gene/protein/metabolites for our analysis and quantification. Multi-omics platforms can merge with events in clinical part, including treatment options, drug response or clinicopathological factors. From these data, reliable prognostic and diagnostic factors can be obtained, and the results can also be judged by the blood, urine, bile as well as the primary tumor of CCA patients. On the other side, the combination of omics-based dataset and bioinformatics tools can reveal available/underlying signatures and mechanisms. Potential inhibitors/small compounds or drug repurposing can also be used to reverse their observations in precision medicine. Therefore, the omics-based approaches have the potential as useful tools to explore a series of multiplex events in CCA. More important, multi-omics approaches can integrate the latest artificial intelligence and machine-learning to improve the accuracy and remain competitive in future research (Figure 5).

## Figures and Tables

**Figure 1 biomolecules-10-01377-f001:**
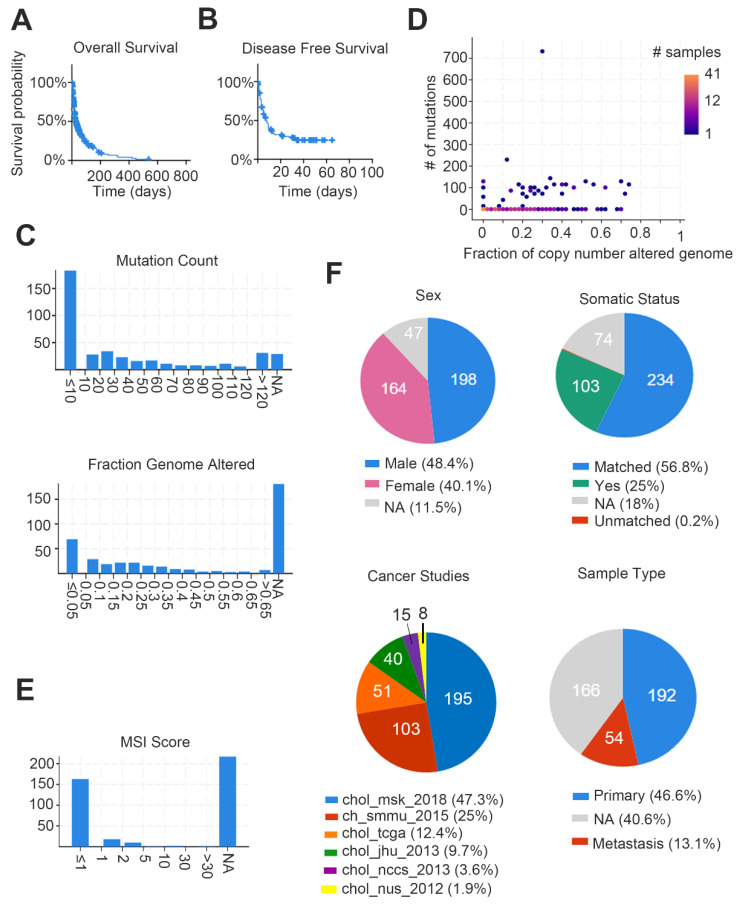
Meta-analysis of cholangiocarcinoma (CCA) demographics and clinical events. Study recruited six cohorts, including MSK, National Cancer Center of Singapore, National University of Singapore, TCGA, JHU and Shanghai) with a total of 409 patients (412 samples). We summarized the clinical information through the cBioPortal website. (**A**) Average overall survival probability in the study; (**B**) average disease-free survival rate in the study; (**C**) ratio of mutation counts and the range of changes in the number of genome alters, Y-axis = samples; (**D**) proportion graph indicates the ratio of mutation counts and the range of changes in the number of genome alters; (**E**) range of microsatellite instability (MSI) scores in the study, Y-axis = samples; (**F**) circle graphs show percentage of various clinical parameters in the study.

**Figure 2 biomolecules-10-01377-f002:**
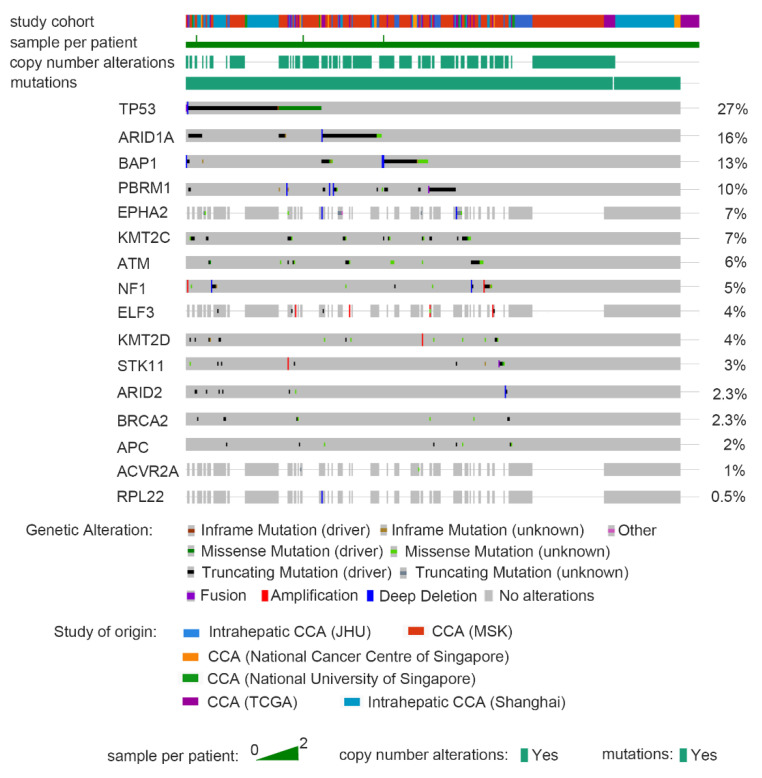
Meta-analysis of genetic alterations in various CCA cohorts. Study recruited six cohorts, including MSK, National Cancer Center of Singapore, National University of Singapore, TCGA, JHU and Shanghai) with a total of 409 patients (412 samples). We summarized the plot by RNA-sequencing of CCA patients on the cBioPortal website. Graph collects the types and patterns of genetic alteration of multiple targets.

**Figure 3 biomolecules-10-01377-f003:**
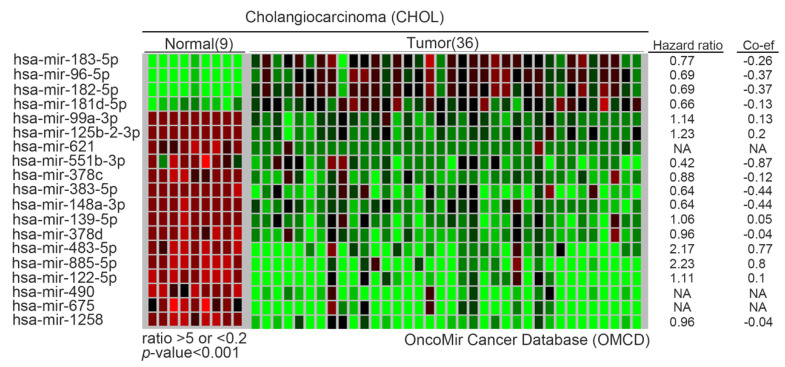
Most significant differences of miRNAs in CCA. List selected from OncoMir Cancer Database (OMCD) website. After statistical analysis (*t*-test), the top four upregulated miRNAs and the top 15 downregulated miRNAs in CCA were selected (ratio > 5 or < 0.2, *p*-value < 0.001).

**Figure 4 biomolecules-10-01377-f004:**
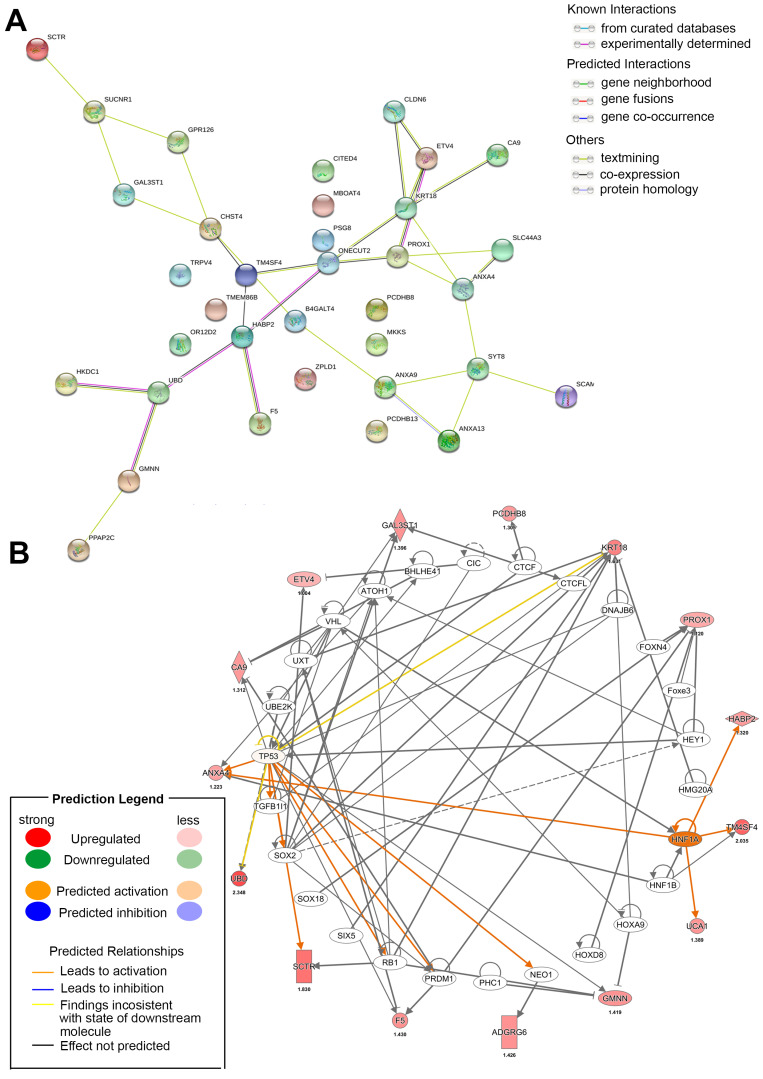
Network of most differential genes in CCA. All candidates were determined that overexpress in CCA tumors compared with normal parts from gene expression profiling interactive analysis (GEPIA) website. (**A**) String and (**B**) ingenuity pathway analysis websites predict potential interactions, transcription factors and networks *(p <* 0.01).

**Figure 5 biomolecules-10-01377-f005:**
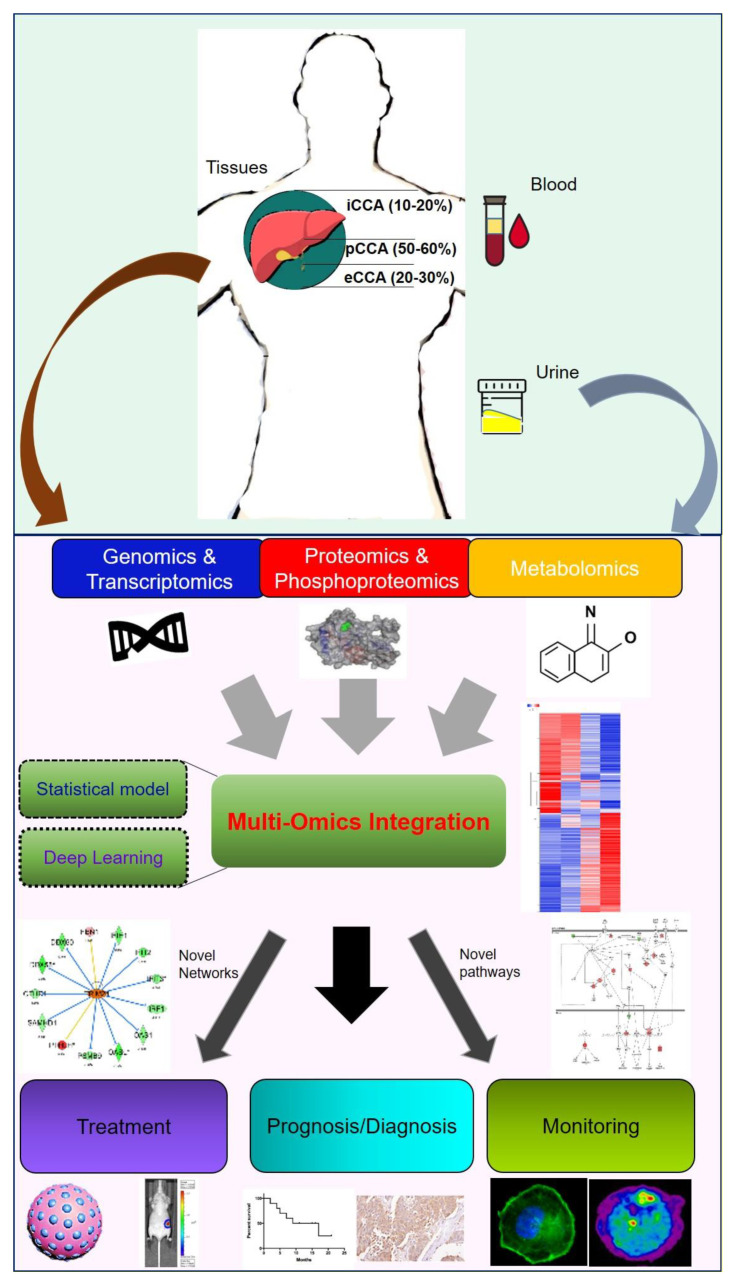
Schematic model of CCA approaches and applications. In this overview, we show several CCA subtypes according to the location of CCA, including intrahepatic (iCCA), perihepatic (pCCA) and extrahepatic (eCCA). We collected previous studies to indicate various mechanisms and experiments in patients’ blood, urine and primary tissue specimens. This schematic model illustrates that omics-based approaches can be used and integrated for further CCA basic and clinical research. In this topic, we input genomics, epigenomics, transcriptomics, proteomics and metabolomics observations for readers.

**Table 1 biomolecules-10-01377-t001:** Genetic events of cholangiocarcinoma (CCA).

**Copy-Number Alteration (CNA)**				
**Gene**	**Cytoband**	**CAN**	**#**	**Frequency (%)**
CDKN2A	9p21.3	HOMDEL	21	9.1
CDKN2B	9p21.3	HOMDEL	16	6.9
MCL1	1q21.2	AMP	11	4.8
CCND1	11q13.3	AMP	8	3.5
FDF4	11q13.3	AMP	8	3.5
MDM2	12q15	AMP	8	3.5
RIT1	1q22	AMP	8	3.5
FDF19	11q13.3	AMP	8	3.5
HIST2H3C	1q21.2	AMP	6	3.4
HIST2H3D	1q21.2	AMP	6	3.4
ERBB2	17q12	AMP	7	3
FGF3	11q13.3	AMP	7	3
MUC1	1q22	AMP	7	3
MYC	8q24.21	AMP	7	3
NOTCH2	1p12	AMP	7	3
**Mutated Genes**				
**Gene**		**#Mut**	**Frequency (%)**	
TP53		114	27.3	
IDH1		64	16.2	
ARID1A		69	16.2	
KRAS		49	12.4	
BAP1		49	11.9	
PBRM1		37	9.3	
KMT2C		28	7.1	
HLA-A		37	6.7	
SMAD4		27	6.3	
ATM		27	6.1	
**Fusion Genes**				
**Gene**		**#Mut**	**Frequency (%)**	
FDFR2		19	4.8	
RAD21		1	0.3	
BRCA1		1	0.3	
BRAF		1	0.3	
ESR1		1	0.3	
MAP3K1		1	0.3	
NOTCH2		1	0.3	
PIK3C2G		1	0.3	
MAP2K1		1	0.3	
RB1		1	0.3	

**Table 2 biomolecules-10-01377-t002:** Most significant miRNAs in CCA is related to survival rate or tumorigenesis. All candidates from WashU Pan-Cancer miRNome Atlas (OncomiR) website. Significance in tumor development was determined through paired Student’s *t*-test, for survival, unpaired Student’s *t*-test and univariate Cox proportional hazards analysis between living and decreased patients.

**miRNAs Significantly Associated with Survival in CHOL**			
**miRNA Name**	**Z-Score**	**Log Rank *p*-Value**	**Upregulated in:**
hsa-miR-802	0.000	3.02 × 10^−3^	Living
hsa-miR-500b-5p	2.718	3.57 × 10^−3^	Living
hsa-miR-500a-5p	2.674	4.27 × 10^−3^	Living
hsa-miR-202-5p	0.000	4.91 × 10^−3^	Living
hsa-miR-551b-3p	1.746	5.57 × 10^−3^	Living
hsa-miR-129-5p	2.166	5.79 × 10^−3^	Living
hsa-miR-3161	3.025	9.51 × 10^−3^	Decreased
hsa-miR-3199	0.173	1.50 × 10^−2^	Living
hsa-miR-1228-5p	0.000	2.03 × 10^−2^	Living
hsa-miR-10b-3p	1.970	2.40 × 10^−2^	Living
**miRNAs Associated with Tumorigenesis of CHOL**			
**miRNA Name**	***t-*** **Test *p*-Value**	***t-*** **Test FDR**	**Upregulated in:**
hsa-miR-183-5p	3.84 × 10^−9^	2.16 × 10^−6^	Tumor
hsa-miR-182-5p	1.15 × 10^−8^	3.25 × 10^−6^	Tumor
hsa-miR-194-3p	2.88 × 10^−8^	5.41 × 10^−6^	Normal
hsa-miR-125b-2-3p	1.35 × 10^−7^	1.85 × 10^−5^	Normal
hsa-let-7c-5p	1.71 × 10^−7^	1.85 × 10^−5^	Normal
hsa-miR-378a-3p	1.97 × 10^−7^	1.85 × 10^−5^	Normal
hsa-miR-92b-3p	2.60 × 10^−7^	2.09 × 10^−5^	Tumor
hsa-miR-1258	3.43 × 10^−7^	2.41 × 10^−5^	Normal
hsa-miR-378c	4.13 × 10^−7^	2.59 × 10^−5^	Normal
hsa-miR-23c	8.50 × 10^−7^	4.36 × 10^−5^	Normal

**Table 3 biomolecules-10-01377-t003:** Most significant LncRNAs/CircRNAs are related to the tumorigenesis of CCA. All candidates from LncRNADisease website (version 2.0).

LncRNAs/CircRNAs Dependency of CCA			
LncRNAs Symbol	Disease Name	Confidence Score	Database ID
SPRY4-IT1	CCA	0.9820	LDA0003741
CPS1-IT1	iCCA	0.9820	LDA0004726
MALAT1	CCA	0.9526	LDA0003742
NEAT1	CCA	0.9526	LDA0003743
SOX2-OT	CCA	0.9462	LDA0003746
AFAP1-AS1	CCA	0.8808	LDA0003736
UCA1	CCA	0.8808	LDA0003749
CCAT1	iCCA	0.8808	LDA0004724
H19	CCA	0.8785	LDA0003740
HULC	CCA	0.8785	LDA0003741
PCAT1	CCA	0.8785	LDA0003745
CCAT2	CCA	0.7311	LDA0003737
PANDAR	CCA	0.7311	LDA0003744
TUG1	CCA	0.7311	LDA0003748
PCAT1	eCCA	0.7311	LDA0004199
TLINC	iCCA	0.7311	LDA0004727
TUG1	iCCA	0.7311	LDA0004728
CDR1-AS	CCA	0.7311	LDA0178708
hsa-circ_0001649	CCA	0.7311	LDA0178709
CCAT2	iCCA	0.6606	LDA0004725
ENST00000517758.1	CCA	0.5483	LDA0003738
ENST00000588480.1	CCA	0.5483	LDA0003739

The confidence scores can be calculated as follows: CS = 1 − $\frac{\Pi }{t}$(1 − $\frac{Wt}{1+e^{-n}}$).

**Table 4 biomolecules-10-01377-t004:** Most significant survival genes in CCA. All candidates from gene expression profiling interactive analysis (GEPIA) website. Log-rank test (Mantel-Cox test), group cutoff: median.

**Overall Survival**		
**Gene Symbol**	**Gene ID**	***p*** **Value**
CTD-2033C11.1	ENSG00000269961.1	4.36 × 10^−5^
AD001527.7	ENSG00000270760.1	1.46 × 10^−4^
FAM86KP	ENSG00000163612.10	4.47 × 10^−4^
RP11-209M4.1	ENSG00000267253.1	9.00 × 10^−4^
PTGER3	ENSG00000050628.20	9.41 × 10^−4^
AP003774.6	ENSG00000231680.1	9.95 × 10^−4^
FAM177A1	ENSG00000151327.12	1.00 × 10^−3^
RP11-574F21.2	ENSG00000228606.1	1.05 × 10^−3^
CXCL17	ENSG00000189377.8	1.14 × 10^−3^
FUT4	ENSG00000196371.3	1.27 × 10^−3^
**Disease-Free Survival**		
**Gene Symbol**	**Gene ID**	***p*** **Value**
AC104654.2	ENSG00000234362.5	1.31 × 10^−6^
AP003774.6	ENSG00000231680.1	5.59 × 10^−6^
NADK	ENSG00000008130.15	3.05 × 10^−5^
AGAP2	ENSG00000135439.11	3.59 × 10^−5^
NRXN2	ENSG00000110076.18	4.80 × 10^−5^
RP1-28O10.1	ENSG00000227591.5	4.81 × 10^−5^
CPSF3L	ENSG00000127054.18	7.45 × 10^−5^
TPRG1L	ENSG00000158109.14	8.32 × 10^−5^
CDC40	ENSG00000168438.14	8.54 × 10^−5^
KLHL34	ENSG00000185915.5	8.86 × 10^−5^

**Table 5 biomolecules-10-01377-t005:** Inhibitors and their related clinical trials for CCA.

Compound	Target	Clinical Trial Number	Phase	Status
NVP-BGJ398 (infigratinib)	FGFR	NCT02150967	II	Recruiting
JNJ-42756493 (Erdafitinib)	FGFR	NCT02699606	II	Recruiting
ARQ 087 (derazantinib)	FGFR	NCT017529520	I + II	Active
TAS-120	FGFR	NCT02052778	I + II	Active
CH5183284 (debio 1347)	FGFR	NCT03834220	II	Active
Ponatinib	FGFR2	NCT02265341	II	Completed
FPA144 (bemarituzumab)	FGFR2	N/A		
Pemazyre (pemigatinib)	FGFR2	NCT03656536	III	Recruiting
AG-120 (Ivosidenib)	IDH1	NCT02073994	I	Active
Trametinib	MEK	NCT02070549/NCT02042443	I/II	Recruiting/Completed
Pazopanib	MEK/VEGFR/PDGFR/REF	NCT01855724	II	Terminated
Pembrolizumab	PD-1	NCT03111732	II	Recruiting
Regorafenib	MEK/VEGFR/PDGFR/REF	NCT02115542	II	Active
AGI-5198	IDH1	N/A		
AGI-6780	IDH2	N/A		
AG-221 (enasidenib)	IDH2	NCT02273739	I + II	Completed
Isoquinoline	PKA	N/A		
S63845	MCL1	N/A		
AZD1480	JAK	N/A		
AZD6244 (selumetinib)	MEK1/2	NCT00553332	II	Completed
BKM120 (buparlisib)	PI3K	NCT01501604	II	Withdrawn
Amatuximab	Mesothelin	NCT01766219	I + II	Completed
Azacitidine	DNMT	N/A		
Decitabine	DNMT	N/A		
Vorinostat	HDAC	N/A		
Romidepsin	HDAC	N/A		
Tivantinib	MET	N/A		
Cabozantinib	MET	NCT01954745	II	Completed

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
