# Peer review of "Omics-Based Platforms: Current Status and Potential Use for Cholangiocarcinoma"

_biomolecules, 2020, doi:10.3390/biom10101377_

Round 1
Reviewer 1 Report
The manuscript entitled “Omics-based platforms: current status and potential use for cholangiocarcinoma” by Chang et al., provides an overview on the main omics based platforms used in cholangiocarcinoma (CCA) researchers , to evaluate to test novel therapeutic approaches for CCA an to identify a new prognostic markers. The authors highlited the crucial fucntion of the multi-omics approaches to overcame the CCA bottlenecks in basic reseach and in clinical application. The manuscript in general is informative, but requires extensive revision. Numerous statements are very general and fail to convey accurate or precise information. The authors should endeavor to be clear, scientifically precise, and accurate.
MAJOR:
- Abstract, Lines 23-34: this part that summarize the main topics treated in the review is confusing, please clarify better and create a better connection.
- Page 3, line 36: Please include the latest WHO pathological classification
made a new categorization of iCCA: small duct and large duct type A small duct CCA is corresponding to mass forming type or/and peripheral type. - In the introduction section, is crucial to introduce the gold standard in the therapy of CCA and describe the huge problem of the chemo resistance.
- In the section 2.1, it would be interesting if the authors include a further studies, since the genomic alteration are studied in depth in CCA pathogenesis. I suggest to the authors to design a table including the studies devoted to identify genomic perturbations.
- In the section 2.2, line 116 include the references for the described study.
- In the section 2.2, line 118 include the references for the described study.
- In the section 2.2, line 124 describe better the role and the activity of DNMT1 in CCA.
- In the section 3, line 151, clarify the role and the effect of MAT1A in CCA.
- In the section 3, line 154, described better the study cited in reference 45
- In the section 3, line 155, what is the role of TNNI3K in CCA?
- In the section 3, line 158, create a better connection between the previously statement and O.Viverrini. It is very nice to create an harmonious text.
- In the section 3, lines 168-172, why do you cite the studies (ref 60,61 and 62) in this section? They regarding the miRNAs, not the lncRNAs and circRNAs, previously described in the section 2.2.
- In the section 3, line 175, the same observation for the study cited in ref 65.
- In the section 4, line 191, describe better the study of Zhang (ref 67). I suppose that the authors identified the proteome profile of mice exposed to 1,2 CDP with the combined technology. You wrote that the authors identified the mice.
- In the section 4, line 199, specify how the authors of the study cited in ref 68 identified the altered protein levels.
- In the section 5, line 210 add a brief description of the multiple candidate proteins and CASK associated with CCA prognosis cited in the study (ref 75).
- In the section 5, line 218, please describe properly the study of Varamo et al. (ref 79).
- In the section 5, line 219, please describe better the linkage MC-LR and iCCA (ref 80).
- In the section 5, line 222, please clarify the intracellular crosstalk between iCCA and vascular CAFs.
- In the section 5, line 224, please add the reference for the study of Saengboonmee and describe the differences between CCA cell line cultures in normal and high glucose medium.
- In the section 6, line 251, explain better the impact of lipids in CCA pathogenesis.
- In the section 8.1, line 289, what are the previous studies, please cite them.
- In the section 8.1, line 304, please mention the newly FDA approved Pemazyre (FGFR2 fusions, 9% to 14% of patients).
- In the section 9, future prospects should be revised. Each sentence should be accurate, precise, and informative. Please elaborate.
MINOR:
- Abstract, Line 20: add molecular before the mechanism
- Abstract, Line 22: add i.e before (TP53 mut, KRAS mut) since they are only an examples of mutations present in CCA.
- Abstract, Line 24: add i.e before (NFKB, YAP1), for the same reason previosly mentioned.
- Section 4, line 202, what is the meaning of the word "festinate"?
- Section 5, line 216 (ref 78), the name of the first author is not correct, please revise.
- Section 6, line 214, please cite here the extended name of IDH1.
- Section 8.2, line 308, please remove issues, is not related in the context.
- Figure 3, the description of the figure is not clear. Please create a better connection and explanation of the schematic illustration.
Author Response
Please see the attachment, Thank you.

Reviewer 2 Report
The authors have provided a timely and comprehensive review on omics data analysis for Cholangiocarcinoma (CCA). The analysis based on multiple types of omics platform has been thoroughly investigated.
My major concern is that the importance of statistical analysis has been severely downplayed, although many arguments from the article have been supported by results from statistical analyses. In general, for analyzing large scale omics data, (robust) variable/feature selection methods are critical in order to identify a small subset of important omics features associated with the disease traits such as cancer prognostic outcomes (Wu and Ma PMID: 25479793, Wu et al. PMC6473252). The effectiveness of (robust) variable selection has been convincingly demonstrated in numerous existing cancer genomics studies. It is an apparent limitation without properly relating these methods to the surveyed studies.
In section 2.1, the authors have conducted analysis by pooling 6 cohorts’ data together. I have to question the validity of the findings as published study has clear shown that pooled cancer survival outcomes from independent studies are heterogeneous, and statistical analysis without accounting for the robustness yields unreliable findings (Ren et al. PMID: 30746793). Especially here, sample sizes from some studies are extremely small (15 and 8) compared to the rest of the studies, which will exacerbate such an issue. It is better to cite the reference to make the limitation of the analysis clear.
Nowadays, integrating multi-omics data for a better understanding of disease progression and etiology has been widely conducted. The review discussed multiple aspects of omics data analysis in detail but failed to identify this important area of research for cancer studies. In particular, Section 7 very vaguely mentioned “multiple-omics approach” but it isn’t about integrating multi-platform omics data. Again, feature selection has been the most widely adopted statistical method for conducting integrative analysis (Wu et al. PMC6473252).
Some additional observations based on the statistical analysis:
In section 2.1, with such a large sample size from 6 cohorts, the most intriguing question is the shortlist of important genes related to CCA survival. So why only investigates the low dimensional clinical covariates? The software package “regnet” associated with Ren et al. PMID: 30746793 can be adopted for a quick analysis if the authors are interested.
Page 3, sub figure F. The sample sizes under sex (409), somatic status (411), cancer studies (412) and sample type (412) are not consistent. Based on the current plot, the category corresponding to 8 is denoted with the same color as the category corresponding to 195.
Table 4: how are these p-values computed? Are they corrected for multiple testing issues?
For the statistical results presented in this study, it might be worthwhile to summarize the corresponding statistical analysis/method/procedure in a small section.
Author Response
Please see the attachment, Thank you.

Round 2
Reviewer 1 Report
The authors partially answered to the issues requested fort the revision. The review needs to be further improved,currently it results generic and inaccurate.
MAJOR:
- In the introduction (lines 45-46), specify the many clinicopathological characteristics of pCCA.
- In the introduction (lines 62-66), as I previously request is crucial to indroduce the first-line therapy for the advanced CCA and the problem of the chemoresistance. The part added is very generic and not significative.
- In section 2.2 (line 147), What is appen after the direct binding of DNMT1 by miR148a and miR152? Please, clarify.
- In section 3 (line 199), decribe in general the role end the function of MAT1A, in this form in not clear.
- In section 5 (lines 282-285), describe better the study of Varamo (ref.22), the current description hot have a sense.
- In section 5 (lines 285-288), clarify the role of MC-LR, is not clear.
- In section 8.1 (lines 395-398), rephrase better the sentences is not correct.
- The future prospects are very unaccetable, please write better this part. It is not possible conclude the review with these random sentence.
- I racommend to perform the English revison with a native speaker. In the manuscript are present many grammar mistakes and also the stylistic form needs to be improved.
MINOR:
- In the abstract section, replace "ex" with "i.e"
- In the abstract section (line 23), replace "collected" with "rewieved"
- In the introduction (line 65), replace "induce" with "induced
- In section 4 (line 236), remove "do"
- In section 4 (line 246), replace "exposure" with "exposed"
- In section 5 (line 282), replace "training" with "exposure"
- In section 5 (line296), replace "normal" with "standard"
- In section 8.1 (line 379), replace "these"with "aggressive"
- In section 8.1 (line395), replace "blocker" eith "inhibitor"
- In the legend of Figure 5 (line 466), replace "we composed several.." with "we rapresented CCA..."
Author Response
MAJOR:
- In the introduction (lines 45-46), specify the many clinicopathological characteristics of pCCA.
Answer: We have rephrased the sentence :” The small-duct type cancer usually has a history of chronic liver disease (5-year overall survival is longer ~60%, most of which are genetic changes of IDH1, BAP1 and FGFR2), and large-duct iCCA and pCCA have many clinicopathological characteristics and are accompanied by premalignant biliary intraepithelial neoplasia (BiIIN) in adjacent ducts (5-year overall survival is shorter, accounting for about 20%, most of which are genetic changes of SMAD4 and KRAS7)”. (please find lines 42-47 in revised manuscript.)
- In the introduction (lines 62-66), as I previously request is crucial to indroduce the first-line therapy for the advanced CCA and the problem of the chemoresistance. The part added is very generic and not significative.
Answer: we had added the description of 1st line therapy in the introduction part.
For treatment, palliative chemotherapy with gemcitabine and cisplatin is the standard 1st line treatment for unresectable or relapse/recurrence CCA patients. However, the response to such therapy remains limited [22]. Most importantly, CCA can escape several known tolerance mechanisms induced by chemotherapy (including cisplatin, gemcitabine and other combinations)[23]. Drug development that relies on well-known signal transductions needs in-depth discussion. (please find lines 64-69 in revised manuscript.)
- In section 2.2 (line 147), What is appen after the direct binding of DNMT1 by miR148a and miR152? Please, clarify.
Answer: We have rephrased the sentence:” After verification, it was confirmed that the miR-148a/miR-152 family directly binds to the DNMT gene. Interestingly, DNMT1 regulates the hypermethylation of its CpG islands and then loses these tumor suppressor functions. Therefore, they form a negative feedback regulatory loop between DNMT1 and miR-148a/miR-152 family in tumorigenesis.” (please find lines 149-152 in revised manuscript.)
- In section 3 (line 199), decribe in general the role end the function of MAT1A, in this form in not clear.
Answer: We have rephrased the sentence:” MAT1A can mainly synthesize S- adenosylmethionine (SAMe) levels, which can prevent cholestatic liver injury. MAT1A has been identified that the expression level of MAT1A in primary and metastatic CCA is reduced compared with normal bile duct cells.” (please find lines 203-206 in revised manuscript.)
- In section 5 (lines 282-285), describe better the study of Varamo (ref.22), the current description hot have a sense.
Answer: We have rephrased the sentence:” They further predicted several molecular mechanisms through 354 down-regulated and 382 up-regulated targets. These typical pathways include DNA replication, cell cycle and pyrimidine metabolism. In addition, they also found new molecules with differently expressed in their model (QPRT, RRM1, FOSB, PRSS1…ect) [23].” (please find lines 273-277 in revised manuscript.)
- In section 5 (lines 285-288), clarify the role of MC-LR, is not clear.
Answer: We have rephrased the sentence:” MC-LR has recognized that certain hepatoxins can induce liver damage and liver cancer. MC-LR is also correlated with intrahepatic bile duct hyperplasia. Therefore, the group tried to explore the relationship between MC-LR, inflammation and intrahepatic bile duct epithelial cell proliferation.” (please find lines 278-281 in revised manuscript.)
- In section 8.1 (lines 395-398), rephrase better the sentences is not correct.
Answer: We thanks for your suggestion, we rewrite the section 8.1.(please find lines 373-395 in revised manuscript.)
- The future prospects are very unaccetable, please write better this part. It is not possible conclude the review with these random sentence.
Answer: Thanks for your suggestion. We rewrite the “future prospectives” section (please find lines 442-456 in revised manuscript.)
- I racommend to perform the English revison with a native speaker. In the manuscript are present many grammar mistakes and also the stylistic form needs to be improved.
Answer: We thank you for your advice. We will ask the handling editor to evaluate and then send this manuscript again for language editing.
MINOR:
- In the abstract section, replace "ex" with "i.e"
- In the abstract section (line 23), replace "collected" with "rewieved"
- In the introduction (line 65), replace "induce" with "induced
- In section 4 (line 236), remove "do"
- In section 4 (line 246), replace "exposure" with "exposed"
- In section 5 (line 282), replace "training" with "exposure"
- In section 5 (line296), replace "normal" with "standard"
- In section 8.1 (line 379), replace "these"with "aggressive"
- In section 8.1 (line395), replace "blocker" eith "inhibitor"
- In the legend of Figure 5 (line 466), replace "we composed several.." with "we rapresented CCA..."
Answer: Thank you for the suggestions from reviewer 1. We have corrected these words in revised manuscript.
Reviewer 2 Report
no further comments.
Author Response
We thank you for your useful suggestions and comments to enhance the value of this article.